# Problem-solving ability and future time perspective among the Chinese nursing interns: The mediating role of future work self

Zhangyi Wang[1]*, Yue Zhu[2], Xiaoping Zhan[3], Tingrui Wang[4], Xiaochun Tang[1], Liping Li[1], Tao Su[5], Huifang Zhou[1], Li Liu[6], Lamei Chen[1], Xiaoli Pang[5], Jiaofeng Peng[7]*, Yan Wang[8]*, Li Yang[9]*

1 Nursing Department, Affiliated Hengyang Hospital of Hunan Normal University & Hengyang Central Hospital, Hengyang, Hunan Province, China, 2 Nursing Department, Tianjin Academy of Traditional Chinese Medicine Affiliated Hospital, Tianjin, China, 3 Children Respiratory Zone 2, Chenzhou No.1 People's Hospital, Chenzhou, Hunan Province, China, 4 School of Nursing, Guizhou Medical University, Guiyang, Guizhou Province, China, 5 School of Nursing, Tianjin University of Traditional Chinese Medicine, Tianjin, China, 6 Nephrology Department / Rheumatology and Immunology Department, The Second Affiliated Hospital of University of South China, Hengyang, Hunan Province, China, 7 Medical Laboratory, Affiliated Hengyang Hospital of Hunan Normal University & Hengyang Central Hospital, Hengyang, Hunan Province, China, 8 Operating Room, The Second Affiliated Hospital of University of South China, Hengyang, Hunan Province, China, 9 Gynecology Department, Chenzhou No.1 People's Hospital, Chenzhou, Hunan Province, China

* 3822785278@qq.com (JP); 25440955@qq.com (YW); 23809392@qq.com (LY); 283537548@qq.com (ZW)

**Data Availability Statement:** All relevant data are within the paper and its Supporting information files.

## Abstract

### Background

The significance of problem-solving ability has been confirmed in numerous studies world-wide, highlighting its role in enhancing the skills of nursing interns and reducing psychological pressure. However, existing research indicates that the problem-solving ability of nursing interns urgently needs to be further improved. Limited research has been conducted on the problem-solving ability of nursing interns, and the correlations among problem-solving ability, future time perspective, and future work self of Chinese nursing interns are unclear.

### Objectives

To investigate problem-solving ability, future time perspective, and future work self among the Chinese nursing interns, and to examine the relationships among these variables. Additionally, the study aims to explore the mediating role of future work self between problem-solving ability and future time perspective.

### Methods

A cross-sectional and correlational design was employed, adhering to the quality reporting conformed to the STROBE Checklist. From May 8, 2023, to February 15, 2024, 1,251 nursing interns were recruited from 15 tertiary grade-A hospitals across six cities in China. The Demographic Characteristics Questionnaire, Social Problem-Solving Inventory, Future

**Funding:** This study was supported by the Tianjin Research Innovation Project for Postgraduate Students (CN) [grant numbers: 2021YJSS171], the Health Research Project of Hunan Provincial Health Commission (CN) [grant numbers: W20243278], the Tianjin University of Traditional Chinese Medicine Science and Technology Innovation Fund Project for College Students (CN) [grant numbers: ZX01], and the Hengyang Science and Technology Plan Project (CN) [grant numbers: 202222035776]. The funders of Zhangyi Wang, Liping Li, Yue Zhu, and Xiaochun Tang had role in study design, data collection and analysis, decision to publish, and preparation of the manuscript.

**Competing interests:** The authors have declared that no competing interests exist.

Time Perspective Inventory, and Future Work Self Scale were used. The data was analyzed using descriptive statistics, univariate, correlation, and process plug-in mediation effect analyses.

## Results

The total scores for problem-solving ability, future time perspective, and future work self were 64.39 ± 18.55, 45.08 ± 11.37, and 16.92 ± 5.28, respectively. Problem-solving ability was positively correlated with future time perspective ($r = 0.638$, $p < 0.001$) and future work self ($r = 0.625$, $p < 0.001$). Additionally, future work self partially mediated mediating role between problem-solving ability and future time perspective, accounting for 39.7% of the total effect.

## Conclusion

The problem-solving ability, future time perspective, and future work self among the Chinese nursing interns were relatively moderate, indicating a need for improvement. It is suggested that nursing managers and educators should actively implement career management and planning programs. By enhancing the future time perspective and future work self of nursing interns, their problem-solving ability can be improved. This, in turn, will facilitate their adaptation to clinical work, enhance the quality of nursing care, and promote the development of their nursing profession.

## 1. Introduction

With the development of modern medicine and the prevalence of a holistic nursing service model encompassing mind-body-social-spirit aspects, patients' demands for nursing care are increasing. This requires nurses to possess higher comprehensive quality, theoretical knowledge, and clinical practice skills, such as problem-solving, scientific research, and teaching skills, as well as a solid educational background. Consequently, nurses' roles and responsibilities are becoming more challenging [1]. Nursing students, as the reserve force of the nursing profession, often lack rational thinking and decision-making abilities when identifying patients' various health problems. This deficiency is due to their changing environment and role, unskilled nursing techniques, and limited clinical experience. These factors easily lead to anxiety and psychological pressure, affecting their professional adaptation and development. At this time, nursing students are about to transition from being students to new nurses. Due to their inadequate problem-solving abilities, they may encounter various issues when facing complex and dynamic work pressures and transformation challenges. This situation is not conducive to stabilizing nursing talent and improving nursing quality [2–4].

Problem-solving ability is defined as the capacity to identify complex problems, develop effective and targeted solutions, implement them, and address the root causes of these problems. It is one of the essential competencies in a nurse's career [5,6]. This ability helps individuals to identify problems and find solutions, and those with strong problem-solving abilities actively seek out resources to solve problems and adapt to unexpected situations [7]. In the nursing profession, problem-solving ability significantly impacts nurses' performance, the quality of care, and patient health outcomes [7]. Nurses with high problem-solving abilities are more likely to establish harmonious and cordial interpersonal relationships in the work

environment and better handle patients' problems [5]. Furthermore, problem-solving ability helps to improve various abilities of nursing interns and reduce psychological stress [8]. However, several studies have shown that the problem-solving abilities of Chinese nursing students, especially nursing interns, are not at an optimal level and require significant improvement [9–11]. Therefore, nursing interns must develop problem-solving skills to improve their actual clinical ability, This will promote the personal development of nursing interns and improve the quality of nursing services and patient satisfaction.

However, existing research mainly focuses on the current state of problem-solving abilities among nursing students and newly recruited nurses, along with analyses of general demographic characteristics. There is a notable lack of research on the problem-solving abilities of nursing interns and the exploration of related correlational and mediating roles in China. Therefore, it is essential to investigate the problem-solving abilities of Chinese nursing interns and explore the relationships and mediating roles between these abilities and other related variables.

## 1.1. Literature review

**Future time perspective can positively predict problem-solving ability.** Future time perspective refers to an individual's self-perception, emotional experience, and direction of action regarding their potential for development, ultimately driving the individual to behave in ways that align with their future short- or long-term goals [12]. This perspective helps nursing interns better face the future, plan for it, and act accordingly, significantly impacting their future. Nursing interns who are more concerned about their future development are likely to have clearer plans and are more proactive in shaping their future. Such individuals are less afraid of difficulties, more rational and methodical, and more committed to achieving their goals. This approach leads to continuous improvement in their problem-solving abilities and motivation at work. However, research has shown that the future time perspective of Chinese nursing interns is currently at a poor level and needs to be further improved [13].

**Future work self is a positive predictor of problem-solving ability.** Strauss et al. [14] first introduced the concept of the future work self, which is the degree to which an individual's self-perception can imagine their future work with clarity. It was found that future work self can help individuals to better set career goals and clarify the behavioral paths they need to take [15]. According to self-regulation theory, the clearer an individual is about themselves, the more effectively their self-regulatory system can drive them towards their goals, resulting in more practical actions [16]. A positive future work self serves as a motivating factor, pushing individuals to pursue better performance at work [14]. Individuals with good future work self will put in more effort to reach their goals, thereby bridging the gap between their current reality and their future self at work [17]. Therefore, nursing interns with greater clarity of their future work self are likely to have greater problem-solving abilities. However, research has shown that the future work self of Chinese nursing interns is at a poor level and needs to be further improved [17].

**Future work self may play a mediating role between problem-solving ability and future time perspective.** Individuals with a strong future time perspective are more focused on the future, anticipate future needs, and envision and plan their future work, a cognitive process that promotes a clearer imagination of their future work self-image and helps them develop a well-defined future work self [14]. Furthermore, when individuals have a well-defined future work self, they are more positive about their current work and are more likely to improve their problem-solving ability [18]. Thus, future work self may play a mediating role between problem-solving ability and future time perspective.

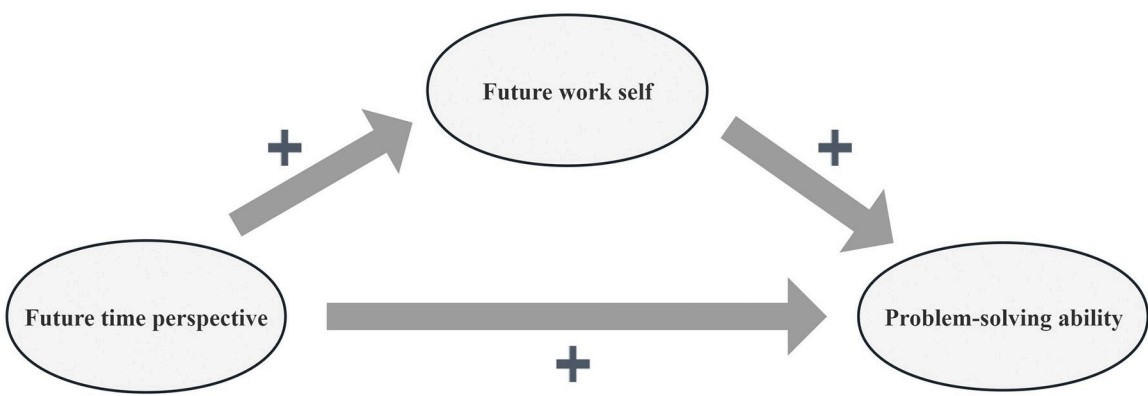

**Fig 1. The conceptual framework of problem-solving ability, future time perspective, and future work self of the study.**

In summary, future time perspective and future work self may have a positive predictive effect on the problem-solving ability of nursing interns, and future work self may play a mediating role between problem-solving ability and future time perspective. However, existing research has primarily focused on the effects of variables such as critical thinking, self-directed learning, and psychological resilience on problem-solving ability. There has been no research on the correlation between problem-solving ability, future time perspective, and future work self among Chinese nursing interns. Thus, the correlations between problem-solving ability, future time perspective, future work self, and their mediating roles in the Chinese nursing interns remain open for further research. Based on the literature review and the aforementioned theory, the conceptual framework of this study was constructed, as shown in **Fig 1**.

## 2. Objectives

This study aimed to (1) investigate the problem-solving ability, future time perspective, and future work self among Chinese nursing interns; (2) examine the correlations between problem-solving ability, future time perspective, and future work self; (3) explore the mediating role of future work self between problem-solving ability and future time perspective; and (4) provide a theoretical basis for constructing the interventional measures to improve the problem-solving ability of the Chinese nursing interns.

## 3. Methods

### 3.1. Study design

This study used a descriptive cross-sectional design and was conducted in China. The quality reporting of the study adhered to the Strengthening the Reporting of Observational Studies in Epidemiology (STROBE) guidelines (see S1 File), which outline the necessary information to include in reports of cross-sectional studies.

### 3.2. Participants and settings

Convenience sampling was used to recruit nursing interns from 15 tertiary grade-A hospitals across six cities in China: Tianjin and Shijiazhuang in the north, and Guiyang, Changsha, Chenzhou, and Hengyang in the south. The recruitment period was from May 8, 2023, to February 15, 2024. Respondents met the following inclusion criteria: (1) clinical internship nursing students in their post; (2) internship duration ≥ 1 month; and (3) voluntary participation

with written informed consent obtained. Those who were on leave or a leave of absence and those with serious physical and psychological illnesses were excluded from this study.

G*Power 3.1 software was used to calculate the minimum sample size required for correlation analysis (α error probability was 0.05 and test validity was 80%). According to related research, the average correlation coefficient among problem-solving ability, future time perspective, and future work self is 0.16 [19–21], and the sample size was calculated using a *t*-test. In G*Power 3.1 software, the *t*-tests method was selected, and "Correlation: Point biserial model" was selected. The parameters were set to an effect size $|\rho| = 0.16$, $\alpha = 0.05$, and $\beta = 0.80$, resulting in a calculated total sample size of 237, which was the minimum sample size for this study. Ultimately, 1,251 participants were included in this study.

## 3.3. Measurements

**3.3.1. The demographic characteristics questionnaire.** The Demographic Characteristics Questionnaire was developed and used to investigate 14 demographic characteristics of Chinese nursing interns. These characteristics included gender, age, educational background, residence, character traits, whether the intern is an only child, and whether the intern holds a student cadre position, among others.

**3.3.2. The Social Problem-Solving Inventory (SPSI).** The Social Problem-Solving Inventory (SPSI) was used to assess the problem-solving ability of nursing interns. The scale, compiled by Wang et al. [19], comprises 25 items across 5 dimensions: "Negative problem orientation" (5 items), "Rational problem-solving" (5 items), "Positive problem orientation" (5 items), "Avoidance style" (6 items), and "Impulsive/negligent style" (4 items). Responses are scored using a Likert scale ranging from 1 (not at all) to 5 (very), with total scores ranging from 5 to 125 points. Higher scores indicate higher problem-solving ability. The scale demonstrated high internal consistency, with a Cronbach's α of 0.925 in Wang et al.'s study and 0.963 in the current study.

**3.3.3. The Future Time Perspective Inventory (FTPI).** The Future Time Perspective Inventory (FTPI), developed by Song et al. [20], was used to assess the future time perspective of nursing interns. The scale comprises 5 dimensions and 20 items: "Long-term target orientation" (5 items), "Future intentions" (4 items), "Behavioral commitments" (4 items), "Future effectiveness" (3 items), and "Awareness of purpose" (4 items). Responses are scored using a Likert scale ranging from 1 (not at all) to 4 (fully). The total score ranges from 20–80 points, with higher scores indicating a better future time perspective. The scale demonstrated high internal consistency, with a Cronbach's α of 0.935 in Song et al.'s study and 0.928 in the current study.

**3.3.4. The Future Work Self Scale (FWSS).** The Future Work Self Scale (FWSS), developed by Guan et al. [21], was used to assess the future work self of nursing interns. The scale consists of 1 dimension and 4 items: "I can easily imagine my future," "This future is easy for me to imagine," "The mental image of this future is very clear," and "I am very clear about who and what I want to be in my future work." Responses are rated on a 7-point Likert scale, ranging from "strongly disagree" to "completely agree." The total score ranges from 4 to 28 points, with higher scores indicating a stronger future work self. The scale demonstrated high internal consistency, with a Cronbach's α of 0.913 in Guan et al.'s study and 0.968 in the current study.

## 3.4. Data collection

Participants were recruited from 15 tertiary grade-A hospitals in six cities across China (Tianjin and Shijiazhuang in the north; Guiyang, Changsha, Chenzhou, and Hengyang in the south) from May 8, 2023, to February 15, 2024. The investigation was conducted with the hospitals'

prior approval, focusing on individual departments and with the assistance of the head nurse in each department for questionnaire distribution. Unified instructions were provided to explain the study's purpose, significance, and confidentiality to participants, who were then surveyed face-to-face using an online questionnaire. Several quality control measures were implemented to ensure the validity of the questionnaires: (1) each participant could fill out the questionnaire only once, as controlled by system settings; (2) participants were required to answer all questions; (3) questionnaires completed in less than one minute were excluded; (4) questionnaires with obviously contradictory answers were excluded (e.g., a 19-year-old with a master's degree, which is implausible for the nursing profession in China); (5) questionnaires with obviously inconsistent personal information were excluded. These measures resulted in the collection of 1,273 questionnaires, of which 1,251 were valid, yielding an effective recovery rate of 98.3%.

## 3.5. Statistical analysis

Two researchers recorded and analyzed the raw data using Epidata 3.1 and IBM SPSS 21.0. Descriptive statistics (numbers and percentage distributions) were used to describe the demographic characteristics. Measurement data following a normal distribution were described using mean ± standard deviation (M ± SD). Group comparisons were performed using two independent sample t-tests or one-way ANOVA for normally distributed data. For non-normally distributed measurement data, the median and interquartile ranges were used, and group comparisons were conducted using Mann–Whitney U or Kruskal–Wallis tests. Pearson's correlation analysis was used to explore correlations between normally distributed variables, while Spearman correlation analysis was used for non-normally distributed variables. The PROCESS plug-in mediation effect analysis was used to investigate the mediating role of future work self between problem-solving ability and future time perspective, with statistical significance set at $p < 0.05$ (two-tailed).

## 3.6. Ethics approval and consent to participate

The study protocol was approved by the Medical Ethics Committee of Hengyang Central Hospital (2023-027-08). Written informed consent was obtained from each participant. All procedures were performed per the principles of the Declaration of Helsinki and relevant guidelines and regulations in China. After obtaining permission from hospital administrators, the researchers approached the participants with the help of the head nurses. The participants were given the right to refuse or withdraw from this study. The questionnaires were designed to ensure anonymity and confidentiality, with no identifying marks, names, or numbers linked to the participants. The data obtained were only used for academic research and not commercial purposes.

# 4. Results

## 4.1. Demographic characteristics

A total of 1,273 Chinese nursing interns were recruited from 15 tertiary grade-A hospitals in six cities across China to participate in the survey. Notably, 22 nursing interns were excluded from the study because of their inconsistent answers, resulting in a final sample of 1,251 nursing interns. Among these participants, 204 (16.3%) were male and 1,047 (83.7%) were female. The age distribution was as follows: 1,030 (82.3%) were aged 19 to less than 22 years old, 189 (15.1%) were aged 22 to less than 24 years old, and 32 (2.6%) were aged 24 years or older. Other demographic characteristics and the results of the univariate analysis are presented in **Table 1**.

**Table 1. Demographic characteristics and univariate analysis of problem-solving ability among the Chinese nursing interns [n = 1,251, M (SD)].**

| Characteristics | n | % | Total score of SPSI | | t / F | p-value |
|---|---|---|---|---|---|---|
| | | | M | SD | | |
| **Gender** | | | | | 0.076 | 0.923 |
| Male | 204 | 16.3 | 64.86 | 22.51 | | |
| Female | 1,047 | 83.7 | 64.15 | 21.96 | | |
| **Age (years old)** | | | | | 15.261 | < 0.001** |
| 19~<22 | 1,030 | 82.3 | 61.28 | 21.09 | | |
| 22~<24 | 189 | 15.1 | 69.63 | 20.56 | | |
| ≥24 | 32 | 2.6 | 78.55 | 22.37 | | |
| **Education background** | | | | | 39.525 | < 0.001** |
| Junior college degree | 199 | 15.9 | 55.29 | 21.55 | | |
| Bachelor degree | 982 | 78.5 | 62.18 | 20.30 | | |
| Master degree or above | 70 | 5.6 | 86.73 | 16.87 | | |
| **Residence** | | | | | 1.582 | 0.127 |
| Cities | 407 | 32.5 | 65.22 | 20.53 | | |
| Township | 508 | 40.6 | 64.31 | 21.01 | | |
| Rural areas | 336 | 26.9 | 63.25 | 20.88 | | |
| **Character trait** | | | | | 2.058 | 0.039 |
| Extroverted | 704 | 56.3 | 67.09 | 21.28 | | |
| Introverted | 547 | 43.7 | 62.38 | 20.59 | | |
| **Whether is a only child** | | | | | 1.378 | 0.171 |
| Yes | 841 | 67.2 | 66.37 | 21.35 | | |
| No | 410 | 32.8 | 63.48 | 21.63 | | |
| **Whether is a student cadre** | | | | | 0.115 | 0.723 |
| Yes | 332 | 26.5 | 65.51 | 20.15 | | |
| No | 919 | 73.5 | 64.17 | 19.86 | | |
| **Preference for nursing profession** | | | | | 28.556 | < 0.001** |
| Dislike | 210 | 16.8 | 51.32 | 19.65 | | |
| General | 779 | 62.3 | 62.18 | 21.52 | | |
| Like | 262 | 20.9 | 73.59 | 20.38 | | |
| **Have been internship time (months)** | | | | | 23.651 | < 0.001** |
| 0~<3 | 158 | 12.6 | 52.86 | 18.26 | | |
| 3~<6 | 942 | 75.3 | 61.25 | 19.58 | | |
| 6~<9 | 98 | 7.8 | 86.12 | 20.31 | | |
| ≥9 | 53 | 4.3 | 89.28 | 21.55 | | |
| **Number of night shifts per month (times)** | | | | | 1.781 | 0.112 |
| <5 | 780 | 62.3 | 63.88 | 21.58 | | |
| 5~10 | 455 | 36.4 | 64.56 | 22.13 | | |
| >10 | 16 | 1.3 | 65.21 | 20.27 | | |
| **Whether are engaged in nursing after graduation** | | | | | 7.883 | < 0.001** |
| Yes | 1,046 | 83.6 | 68.85 | 20.61 | | |
| No | 205 | 16.4 | 51.28 | 21.28 | | |
| **Academic performance during the school** | | | | | 35.625 | < 0.001** |
| Better (top 1/3 of the grade) | 248 | 19.8 | 72.53 | 20.95 | | |
| Intermediate (1/3 of the grade) | 779 | 62.3 | 66.37 | 19.56 | | |
| Poor (bottom 1/3 of the grade) | 224 | 17.9 | 48.22 | 21.28 | | |
| **Satisfaction with clinical teaching** | | | | | 2.528 | 0.065 |
| Dissatisfied | 207 | 16.5 | 62.09 | 21.58 | | |
| General | 729 | 58.3 | 65.28 | 22.13 | | |
| Satisfied | 315 | 25.2 | 67.53 | 20.27 | | |
| **Whether have a plan for future career** | | | | | 7.065 | < 0.001** |

(*Continued*)

**Table 1.** (Continued)

| Characteristics | *n* | % | Total score of SPSI | | *t / F* | *p*-value |
|---|---|---|---|---|---|---|
| | | | **M** | **SD** | | |
| Yes | 792 | 63.3 | 71.54 | 20.75 | | |
| No | 459 | 36.7 | 56.94 | 21.26 | | |

Note
*: $p < 0.05$
**: $p < 0.01$.

### 4.2. Scores of problem-solving ability, future time perspective, and future work self

The problem-solving ability score was 64.39 ± 18.55, and the average SPSI score was 2.58 ± 0.92. Among the dimensions of the SPSI, "Avoidance style" had the highest average score (2.77 ± 0.96), while "Negative problem orientation" had the lowest (2.21 ± 0.89). The average scores of "Rational problem-solving," "Positive problem orientation," and "Impulsive/ negligent style" were 2.69 ± 0.93, 2.62 ± 0.95, and 2.53 ± 0.91, respectively.

The total score for the future time perspective was 45.08 ± 11.37, and the average FTPI score was 2.25 ± 0.63. Among the five dimensions of the FTPI, "Awareness of purpose" had the highest average score (2.42 ± 0.93), while "Future effectiveness" had the lowest (2.11 ± 0.58). The average scores for "Long-term target orientation," "Behavioral commitments," and "Future intentions" were 2.31 ± 0.65, 2.22 ± 0.70, and 2.16 ± 0.62, respectively.

The total score for future work self was 16.92 ± 5.28, and the average FWSS score was 4.23 ± 1.27. The SPSI, FTPI, and FWSS scores are presented in **Table 2**.

### 4.3. Relationships between problem-solving ability, future time perspective, and future work self

The total problem-solving ability score was positively correlated with the total future time perspective score ($r = 0.638$, $p < 0.01$), with positive correlations observed across all dimensions

**Table 2. The scores of SPSI, FTPI, and FWSS among the Chinese nursing interns [*n* = 1,251, M (SD)].**

| Dimensions | Number of items | Dimensional score | | Average score of items | | Ranking |
|---|---|---|---|---|---|---|
| | | **M** | **SD** | **M** | **SD** | |
| **SPSI total score** | 25 | 64.39 | 18.55 | 2.58 | 0.92 | — |
| Negative problem orientation | 5 | 11.05 | 4.03 | 2.21 | 0.89 | 5 |
| Rational problem-solving | 5 | 13.45 | 4.96 | 2.69 | 0.93 | 2 |
| Positive problem orientation | 5 | 13.15 | 4.88 | 2.62 | 0.95 | 3 |
| Avoidance style | 6 | 16.62 | 5.32 | 2.77 | 0.96 | 1 |
| Impulsive/negligent style | 4 | 10.12 | 3.61 | 2.53 | 0.91 | 4 |
| **FTPI total score** | 20 | 45.08 | 11.37 | 2.25 | 0.63 | — |
| Long-term target orientation | 5 | 11.55 | 3.01 | 2.31 | 0.65 | 2 |
| Future intentions | 4 | 8.64 | 2.63 | 2.16 | 0.62 | 4 |
| Behavioral commitments | 4 | 8.88 | 2.58 | 2.22 | 0.70 | 3 |
| Future effectiveness | 3 | 6.33 | 2.25 | 2.11 | 0.58 | 5 |
| Awareness of purpose | 4 | 9.68 | 2.73 | 2.42 | 0.73 | 1 |
| **FWSS total score** | 4 | 16.92 | 5.28 | 4.23 | 1.27 | — |

**Table 3. The relationships between problem-solving ability, future time perspective, and future work self among the Chinese nursing interns ($n$ = 1,251, r).**

| Item | 1 | 1.1 | 1.2 | 1.3 | 1.4 | 1.5 | 2 | 2.1 | 2.2 | 2.3 | 2.4 | 2.5 | 3 |
|---|---|---|---|---|---|---|---|---|---|---|---|---|---|
| **1 SPSI total score** | — | | | | | | | | | | | | |
| 1.1 Negative problem orientation | 0.907** | — | | | | | | | | | | | |
| 1.2 Rational problem-solving | 0.903** | 0.889** | — | | | | | | | | | | |
| 1.3 Positive problem orientation | 0.893** | 0.883** | 0.879** | — | | | | | | | | | |
| 1.4 Avoidance style | 0.885** | 0.874** | 0.871** | 0.873** | — | | | | | | | | |
| 1.5 Impulsive/negligent style | 0.872** | 0.866** | 0.865** | 0.859** | 0.864** | — | | | | | | | |
| **2 FTPI total score** | 0.638** | 0.632** | 0.629** | 0.622** | 0.631** | 0.625** | — | | | | | | |
| 2.1 Long-term target orientation | 0.613** | 0.603** | 0.605** | 0.611** | 0.622** | 0.618** | 0.865** | — | | | | | |
| 2.2 Future intentions | 0.627** | 0.620** | 0.636** | 0.621** | 0.628** | 0.616** | 0.863** | 0.855** | — | | | | |
| 2.3 Behavioral commitments | 0.634** | 0.632** | 0.641** | 0.623** | 0.631** | 0.631** | 0.849** | 0.839** | 0.851** | — | | | |
| 2.4 Future effectiveness | 0.628** | 0.623** | 0.625** | 0.627** | 0.637** | 0.642** | 0.852** | 0.834** | 0.832** | 0.862** | — | | |
| 2.5 Awareness of purpose | 0.625** | 0.629** | 0.626** | 0.621** | 0.618** | 0.613** | 0.823** | 0.827** | 0.823** | 0.841** | 0.846** | — | |
| **3 FWSS total score** | 0.625** | 0.621** | 0.612** | 0.596** | 0.613** | 0.586** | 0.609** | 0.611** | 0.598** | 0.603** | 0.605** | 0.595** | — |

Note

**: $p < 0.01$, —: $r = 1$.

($r = 0.603–0.642$, $p < 0.01$). Additionally, the total problem-solving ability score was positively correlated with the total score for future work self ($r = 0.625$, $p < 0.01$), with positive correlations also observed across all dimensions ($r = 0.586–0.621$, $p < 0.01$), as shown in **Table 3**.

## 4.4. Mediating effect of future work self between problem-solving ability and future time perspective

The total impact of future time perspective on problem-solving ability was 0.451 ($p < 0.01$), with a 95% confidence interval (CI) of 0.225–0.512. The direct effect of future time perspective on problem-solving ability was 0.272 ($p < 0.01$), with a 95% CI of 0.208–0.483. The indirect effect of future work self on problem-solving ability was calculated as 0.573×0.312 = 0.179, accounting for 39.7% of the total effect value of 0.451 ($p < 0.01$). The bootstrapped CI ranged from 0.136 to 0.288, which, excluding 0, indicates that the difference was statistically significant ($p < 0.05$), as shown in **Table 4** and **Fig 2**.

**Table 4. The mediating effect of future work self between problem-solving ability and future time perspective among the Chinese nursing interns ($n$ = 1,251).**

| Model pathways | Standardized effect ($B$) | SE | $t$-value | $p$-value | F | R | $R^2$ | 95% Cl |
|---|---|---|---|---|---|---|---|---|
| **Total effect** | | | | | 386.52 | 0.612 | 0.375 | |
| Future time perspective → Problem-solving ability | 0.451 | 0.035 | 12.886 | < 0.001** | | | | [0.225, 0.512] |
| **Direct effect** | | | | | 213.86 | 0.536 | 0.287 | |
| Future time perspective → Future work self | 0.573 | 0.047 | 12.191 | < 0.001** | | | | [0.336, 0.531] |
| Future time perspective → Problem-solving ability | 0.272 | 0.023 | 11.826 | < 0.001** | | | | [0.208, 0.483] |
| Future work self → Problem-solving ability | 0.312 | 0.028 | 11.143 | < 0.001** | | | | [0.193, 0.312] |
| **Indirect effect** | | | | | — | — | — | |
| Future time perspective → Future work self → Problem-solving ability | 0.179 | 0.015 | — | — | | | | [0.136, 0.288] |

Note

**: $p < 0.01$.

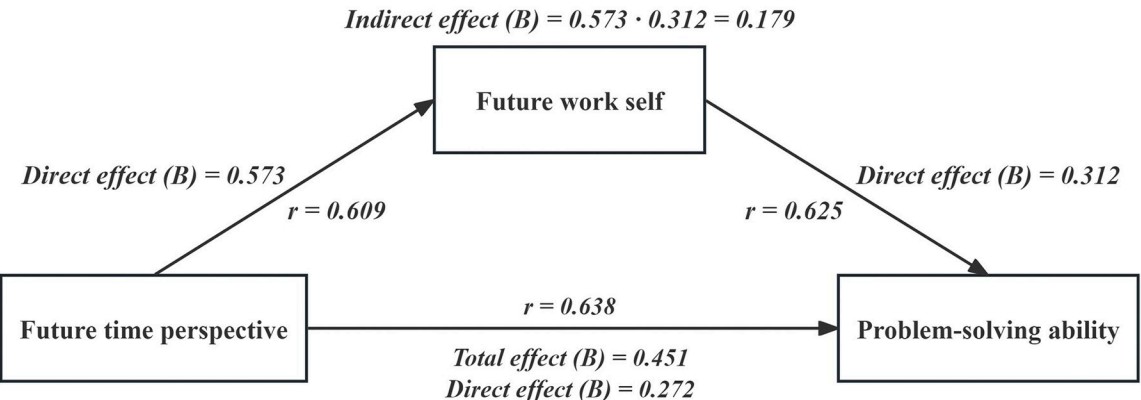

Fig 2. The direct, indirect, and total effects between the problem-solving ability, future time perspective, and future work self.

## 5. Discussion

### 5.1. Status quo of the problem-solving ability, future time perspective, and future work self

The total score for problem-solving ability among the 1,251 Chinese nursing interns was 64.39 ± 18.55. The average score was 2.58 ± 0.92, which is relatively low compared to the median score of 75.00, similar to the findings of Yalin et al. [10]. "Avoidance style" had the highest score, followed by "Rational problem solving", "Positive problem orientation", and "Impulsive/negligent". This finding indicates that while most nursing students view problems as challenges and follow the principles and strategies of effective problem-solving to solve them rationally and systematically, they tend to procrastinate, depend on others, be passive, inactive, panic, and become distracted, leading to incomplete problem resolution. The reasons for this may be that nursing interns face a change of environment and role after entering clinical practice, and their lack of clinical experience, inexperience in nursing practice, poor ability to deal with emergencies, and tensions in the nurse-patient relationship make them prone to adopt an avoidance style when solving problems [22]. As future leaders in nursing, the problem-solving ability of nursing interns is crucial for improving the quality of nursing care and promoting the development of the nursing profession [23]. Therefore, it is suggested that schools and clinical instructors focus on the cultivation of the problem-solving ability of nursing interns, actively conduct training programs on problem-solving ability, and promote the improvement of the clinical nursing ability of nursing interns using flipped classrooms, nursing check-ups, and nursing ability competition for nursing interns. Such initiatives can improve the problem-solving ability of nursing interns.

The total future time perspective score for the 1,251 Chinese nursing interns was 45.08 ± 11.37, with an average score of 2.25 ± 0.63, indicating an intermediate level. This score is lower than that reported by Liu et al. [24]. Among the dimensions, the highest score was for the "Awareness of purpose" dimension, indicating that nursing interns are concerned about their future development and have a clear understanding of their future. The lowest score was for the "Future effectiveness" dimension, indicating a lack of confidence among nursing interns in their ability to create a better future. This may be due to a lack of work experience, poor communication ability with patients, and a tendency to feel overwhelmed in the face of unexpected situations, as well as a lack of awareness of the values of the nursing profession and a general perception that nursing has a heavy workload and low social status. Future time perspective is a value belief that leads to the formation of future goals, which can facilitate

continuous self-adjustment by nursing interns and lead to changes in their cognition and behavior [24]. Therefore, it is suggested that schools and hospital lead teachers emphasize the importance of professional values education. This can be achieved by discussing the meaning, importance, and development prospects of the nursing profession through additional professional quality elective courses and role model education. Such initiatives aim to enhance nursing interns' confidence in the nursing profession and enable them to approach problems during their internships with a positive mindset.

The total score of future work self for the 1,251 Chinese nursing interns was 16.92 ± 5.28, with an average score of 4.23 ± 1.27, also at an intermediate level. This score is lower than the results reported by Fris et al. [25] study on medical students, indicating that nursing interns' perceptions of their future work development and goals were not clear enough. The reasons for this may be that: (1) career planning education in higher education institutions in China is still in its infancy and fails to meet the needs of practicing nurses [26] and (2) nursing students tend to feel confused about their future career development due to heavy tasks and high learning pressure during their internship. Research has shown that future work self can motivate individuals to engage in more proactive career behaviors [14]. Therefore, it is suggested that schools and hospital instructors focus on cultivating the career planning abilities of nursing interns. This can be accomplished by offering career planning courses and organizing activities such as presentations of outstanding nurses' achievements and Satya group counseling sessions to improve nursing interns' future work self and career clarity.

## 5.2. Positive correlations between problem-solving ability, future time perspective, and future work self

A positive correlation was observed between the problem-solving ability of nursing interns and their future time perspective ($r = 0.638$, $p < 0.01$), i.e. the higher the level of future time perspective, the stronger the problem-solving ability of nursing interns, which is consistent with the study of Mulyono et al. [13]. Future time perspective refers to an individual's self-perception, emotional experience, and direction of action regarding potential possibilities for development, and ultimately drives the individual to act in ways that align with their short- and long-term goals [12]. It has been argued that more future-oriented individuals tend to have greater problem-solving abilities, as future time perspective can facilitate the development of creative problem-solving skills [27]. According to cognitive-motivational theory, future time perspective originates from motivational goals and is shaped by a cognitive outlook that allows individuals to predict the future over a longer period. This perspective enables individuals to plan motivational goals and tasks that guide current behavior, ultimately facilitating the achievement of future goals [28]. Therefore, a high level of future time perspective can motivate nursing interns to grow and progress academically, thereby improving their problem-solving ability. For this reason, it is recommended that nursing educators and managers focus on the development of future time perspective in nursing interns and improve their future time perspective through professional values education, psychological counseling, and training in nurse-patient communication ability, thereby improving their problem-solving ability.

Additionally, the study found that the problem-solving ability of nursing interns was positively correlated with their future work self ($r = 0.625$, $p < 0.01$). Future work self refers to the degree to which an individual's future work self is clear and easy to imagine [14]. Future work self is a motivational resource, encouraging self-improvement and proactive investment in current resources to gain other valuable resources [14]. According to resource conservation theory, individuals are more likely to invest their resources in activities that bring a return on resources, and problem-solving ability significantly impacts the psychological quality and

overall capability of nursing interns, bringing them more resources [29]. As a result, nursing interns with high levels of future work self are more likely to take steps to improve their problem-solving ability to access more resources. For this reason, it is recommended that nurse educators and administrators strengthen career planning education for student nurses and provide opportunities such as mock interviews to increase their confidence in their future work and thus improve their problem-solving ability.

## 5.3. Mediating role of future work self between problem-solving ability and future time perspective

The results demonstrated that future work self partially mediates mediating role between problem-solving ability and future time perspective, with a mediating effect of 39.7% ($p < 0.01$). This suggests that future time perspective not only improves the problem-solving ability of nursing interns but also improves problem-solving ability by increasing future work self. The study showed that individuals with a strong future time perspective tend to be more focused on the future, anticipate future needs, and envision and plan future work for themselves, leading to the development of a clear future work self [14]. According to the theory of planned behavior, when nursing interns have a clear future work self, they believe that their current efforts will help them achieve their future work goals. Consequently, they maintain a positive, optimistic, and forward-looking attitude toward their current work and feel more perceptual control over their tasks. Driven by these positive attitudes and perceptual control, nursing interns are likely to be enthusiastic, energetic, and willing to share their knowledge at work. This environment fosters their creative problem-solving abilities [18]. Based on these findings, it is recommended that nursing educators and managers focus on the impact of future time perspective and future work self on interns' problem-solving ability and construct intervention programs to improve nursing interns' problem-solving ability by increasing their future time perspective and future work self.

## 5.4. Strengths and limitations

This study is innovative for several reasons. Firstly, there is limited research on the problem-solving ability of Chinese nursing interns. Secondly, existing research on problem-solving ability has primarily focused on variables such as critical thinking, psychological flexibility, and self-efficacy. This study, however, explores the impact of future time perspective and future work self on the problem-solving ability of nursing interns from the perspective of career planning, which is precisely applicable to nursing interns who are about to enter clinical practice. Additionally, the study suggests that nursing managers and educators should emphasize the development of nursing interns' career planning abilities, and its findings have significant implications for improving the problem-solving abilities of nursing interns.

However, there were some limitations to this study. First, it used a convenience sampling method, enrolling only 1,251 Chinese nursing interns from 15 tertiary grade-A hospitals in six cities across China (Tianjin and Shijiazhuang in the north; Guiyang, Changsha, Chenzhou, and Hengyang in the south). This may result in unrepresentative samples and potentially biased, non-generalizable, or limited findings. Future research should employ random, multi-center sampling with a larger sample size to address this limitation. Second, the study's cross-sectional design may restrict the ability to establish causal relationships. Further longitudinal research is needed to better understand the mechanisms underlying the relationships identified in this study. Third, the study relied solely on an online questionnaire, which might have disadvantages in terms of the accuracy of responses and on-site quality control. Future studies should combine online and offline surveys to reduce potential biases. Lastly, the study did not

employ multiple linear regression analysis to examine the factors influencing problem-solving ability, future time perspective, and future work self. Future research should include a larger number of nursing interns from diverse regions and investigate the factors that influence problem-solving ability more rigorously.

## 6. Conclusion

This study found that the problem-solving ability, future time perspective, and future work self among 1,251 Chinese nursing interns were relatively moderate and needed improvement. In addition, there were significant and positive correlations between problem-solving ability, future time perspective, and future work self. Future work self played a mediating role in the change in problem-solving ability and future time perspective. Based on these findings, it is suggested that nursing managers and educators should actively carry out career management and planning. By improving the future time perspective and future work self of nursing interns, their problem-solving abilities can be enhanced. This will facilitate their adaptation to clinical work, improve the quality of nursing care, and promote the development of nursing careers.

## Supporting information

**S1 File. STROBE checklist.**
(DOCX)

**S2 File. Dataset used for the study.**
(XLSX)

**S1 Appendix. Questionnaires for the study.**
(DOCX)

## Acknowledgments

All researchers would like to express our gratitude to all the participants for taking their precious time to participate in this study, and also thank the hospital managers and nursing administrators for their strong support and help to this study. Furthermore, we would like to thank KetengEdit (www.ketengedit.com) for its linguistic assistance during the preparation of this manuscript.

## Author Contributions

**Conceptualization:** Zhangyi Wang, Yue Zhu, Li Liu, Lamei Chen, Jiaofeng Peng, Li Yang.

**Data curation:** Zhangyi Wang, Xiaoping Zhan, Xiaochun Tang, Huifang Zhou, Li Yang.

**Formal analysis:** Zhangyi Wang, Xiaoping Zhan, Tingrui Wang, Tao Su, Jiaofeng Peng, Li Yang.

**Funding acquisition:** Zhangyi Wang, Yue Zhu, Xiaochun Tang, Liping Li, Li Liu.

**Investigation:** Zhangyi Wang, Yue Zhu, Xiaoping Zhan, Tingrui Wang, Liping Li, Tao Su, Huifang Zhou, Li Liu, Jiaofeng Peng.

**Methodology:** Zhangyi Wang, Yue Zhu, Xiaoping Zhan, Liping Li, Li Liu, Xiaoli Pang, Jiaofeng Peng, Yan Wang, Li Yang.

**Project administration:** Zhangyi Wang, Yue Zhu, Xiaochun Tang, Liping Li, Xiaoli Pang.

**Resources:** Tao Su, Huifang Zhou, Lamei Chen, Yan Wang.

**Validation:** Tingrui Wang, Lamei Chen, Yan Wang.

**Visualization:** Tao Su, Xiaoli Pang.

**Writing – original draft:** Zhangyi Wang, Yue Zhu, Xiaoping Zhan, Tingrui Wang, Li Liu, Yan Wang, Li Yang.

**Writing – review & editing:** Zhangyi Wang, Xiaoping Zhan, Xiaochun Tang, Li Liu, Xiaoli Pang.

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
