## [Decision Letter · Decision Letter 0]

14 Jun 2024

PONE-D-24-16452Correlation between problem-solving ability and future time perspective among the Chinese nursing interns: The mediating role of future work selfPLOS ONE

Dear Dr. Wang,

Thank you for submitting your manuscript to PLOS ONE. After careful consideration, we feel that it has merit but does not fully meet PLOS ONE’s publication criteria as it currently stands. Therefore, we invite you to submit a revised version of the manuscript that addresses the points raised during the review process.

We look forward to receiving your revised manuscript.

Kind regards,

Mukhtiar Baig, Ph.D.

Academic Editor

PLOS ONE

Reviewers' comments:

Reviewer's Responses to Questions

**Comments to the Author**

1. Is the manuscript technically sound, and do the data support the conclusions?

Reviewer #1: Yes

Reviewer #2: Yes

2. Has the statistical analysis been performed appropriately and rigorously? 

Reviewer #1: Yes

Reviewer #2: Yes

3. Have the authors made all data underlying the findings in their manuscript fully available?

Reviewer #1: Yes

Reviewer #2: Yes

4. Is the manuscript presented in an intelligible fashion and written in standard English?

Reviewer #1: No

Reviewer #2: Yes

5. Review Comments to the Author

Reviewer #1: The study titled "Correlation between problem-solving ability and future time perspective among the Chinese nursing interns: The mediating role of future work self" aims to explore correlations among problem-solving ability, future time perspective, and future work self of Chinese nursing interns, and

The study was conducted using a descriptive, cross-sectional design in eight hospitals China.

Overall, this study provides valuable insights into the experiences of nursing interns and highlights the importance of factors such as problem-solving ability and future time perspective. However, it is worth noting that the study has a cross-sectional design, which limits the ability to establish causal relationships. Further longitudinal research is needed to better understand the mechanisms underlying the relationships identified in this study.

Furthermore, the study relies on self-report measures from 15 tertiary grade-A hospitals in China. However, the author did not describe the geographical location of the 15 hospitals in detail, so I have reason to doubt whether this study can represent the whole situation in China. Because the geographic location can bias the results.

In addition, there are some problems in the format and language of the article, please modify it according to the magazine guide.1. The background part needs to be streamlined and too long. It does not briefly summarize the current situation of the future time insight and problem solving ability of Chinese interns. It is just a simple description, which I think is not desirable.2. Some of the expressions of the article are a little stiff, and I think the author needs to modify the language.

Reviewer #2: The definition of "future time perspective" is not outlined at the beginning of the introduction section; it would be best to include it in the beginning to make reading it easier.

You specified that "questionnaires with obvious contradictory answers were excluded"; I believe that detailing the criteria would add to the robustness of the methodology.

6. PLOS authors have the option to publish the peer review history of their article (what does this mean?). If published, this will include your full peer review and any attached files.

Reviewer #1: No

Reviewer #2: No

---

## [Author Response · Author response to Decision Letter 0]

21 Jun 2024

Manuscript Number: PONE-D-24-16452 R1

Title: Problem-solving ability and future time perspective among the Chinese nursing interns: The mediating role of future work self 

June 21th, 2024

Dear academic editor and reviewers:

 We would like to thank you for your efforts in reviewing our revised manuscript entitled “Problem-solving ability and future time perspective among the Chinese nursing interns: The mediating role of future work self”, and providing many helpful comments and suggestions, which will all prove invaluable in the revision and improvement of our paper, as well as in guiding our research in the future.

 According to your nice suggestions, we have studied your comments point-by-point, and have made extensive corrections to our previous draft, and all the changes have been marked in red in the revised draft, the detailed corrections are listed below. In order for reviewers and readers to better understand and read, we invited two nursing professors whose mother tongue is English and KetengEdit (www.ketengedit.com) to polish and revise the language of our article. All authors have approved the response letter and the revised version of the manuscript. 

Responds to the reviewers’ comments point-by-point:

Response to reviewer # 1: 

 Dear reviewer # 1, we appreciate it very much for these good suggestion. We are well aware that there are still some shortcomings in the manuscript. Therefore, in accordance with your suggestions, we have made the following revisions to our manuscript: 

 Q1: “However, it is worth noting that the study has a cross-sectional design, which limits the ability to establish causal relationships. Further longitudinal research is needed to better understand the mechanisms underlying the relationships identified in this study.”

 Response: Thank you very much for your valuable comments, for this suggestion, we are sorry and want to explain that due to the limitation of project funds, time and research conditions, so far, our research group has not carried out longitudinal research, which is indeed a major deficiency of this study, and we have written it in the section of “5.4. Strengths and limitations”, that is “Second, the study's cross-sectional design may restrict the ability to establish causal relationships. Further longitudinal research is needed to better understand the mechanisms underlying the relationships identified in this study.” Thus, longitudinal research will continue to be carried out to better explore the mechanism in the future.

 And all the changes have been marked in red in the “Revised Manuscript with Track Changes”. We hope you can understand and be generous.

 Q2: “Furthermore, the study relies on self-report measures from 15 tertiary grade-A hospitals in China. However, the author did not describe the geographical location of the 15 hospitals in detail, so I have reason to doubt whether this study can represent the whole situation in China. Because the geographic location can bias the results.”

 Response: Thank you very much for your valuable comments, for this suggestion, we have made some major revisions, that is: We have added the cities and geographical regions where 15 tertiary grade-A hospitals are located, that is “Convenience sampling was used to recruit nursing interns from 15 tertiary grade-A hospitals across six cities in China: Tianjin and Shijiazhuang in the north, and Guiyang, Changsha, Chenzhou, and Hengyang in the south. The recruitment period was from May 8, 2023, to February 15, 2024.” This may result in unrepresentative samples and potentially biased, non-generalizable, or limited findings. Future research should employ random, multi-center sampling with a larger sample size to address this limitation, which is also one of the limitations of this study, and it is also described in the section of “5.4. Strengths and limitations”.

 And all the changes have been marked in red in the “Revised Manuscript with Track Changes”. We hope you can understand and be generous.

 Q3: “In addition, there are some problems in the format and language of the article, please modify it according to the magazine guide. 1. The background part needs to be streamlined and too long. It does not briefly summarize the current situation of the future time insight and problem solving ability of Chinese interns. It is just a simple description, which I think is not desirable. 2. Some of the expressions of the article are a little stiff, and I think the author needs to modify the language.”

 Response: Thank you very much for your valuable comments, for this suggestion, we have made some revisions, that is: (1) We have modified the format and language according to the magazine guide. In order for reviewers and readers to better understand and read, we invited two nursing professors whose mother tongue is English and KetengEdit (www.ketengedit.com) to polish and revise the language of our article. (2) We have simplified and revised the “Introduction” and combed the framework, and summarized and described the current situation of the problem-solving ability, future time perspective, future work self in the corresponding section of the “1.1. Literature review”. In addition, a subtitle is added to each part of the literature review to make it easier to read and understand, that is : “Future time perspective can positively predict problem-solving ability.” “Future work self is a positive predictor of problem-solving ability.” “Future work self may play a mediating role between problem-solving ability and future time perspective.”

 For other language and content issues of the article, we have also made unified revisions. And all the changes have been marked in red in the “Revised Manuscript with Track Changes”. We hope you can understand and be generous.

Response to reviewer # 2: 

 Dear reviewer # 2, we appreciate it very much for these good suggestion. We are well aware that there are still some shortcomings in the manuscript. Therefore, in accordance with your suggestions, we have made the following revisions to our manuscript: 

 Q1: “The definition of “future time perspective” is not outlined at the beginning of the introduction section; it would be best to include it in the beginning to make reading it easier.”

 Response: Thank you very much for your valuable comments, for this suggestion, we have made some revisions, that is: We have added the definition of “future time perspective” and “future work self ” in the first and second paragraph in the section of the “1.1. Literature review”, that is “Future time perspective refers to an individual's self-perception, emotional experience, and direction of action regarding their potential for development, ultimately driving the individual to behave in ways that align with their future short- or long-term goals”. We also want to explain that the first few paragraphs in the section of the “1. Introduction” are the study background, mainly focusing on the significance and importance of problem-solving ability of the nursing interns and the lack of relevant research methods and contents at present. Therefore, the relationships and mediating role among problem-solving ability, future time perspective, and future work self of the Chinese nursing will be described in detail in the section of the “1.1. Literature review”. And the section “1.1. Literature review” is one of the contents of section “1. Introduction”.

 And all the changes have been marked in red in the “Revised Manuscript with Track Changes”. We hope you can understand and be generous.

 Q2: “You specified that "questionnaires with obvious contradictory answers were excluded"; I believe that detailing the criteria would add to the robustness of the methodology.”

 Response: Thank you very much for your valuable comments, we also believe that detailing the criteria would add to the robustness of the methodology. Thus, for this suggestion, we have made some major revisions, that is: In the section of the “3.4. Data collection”, we have detailed the criteria, that is “(4) questionnaires with obviously contradictory answers were excluded (e.g., a 19-year-old with a master's degree, which is implausible for the nursing profession in China)”

And all the changes have been marked in red in the “Revised Manuscript with Track Changes”. We hope you can understand and be generous.

 Last but not least, we have studied your comments point by point, we tried our best to improve the manuscript and made some changes to the manuscript. These changes will not influence the content and framework of the paper. And here we did not list the changes in detail but marked in red in the revised paper. We appreciate for your warm work earnestly and hope that the correction will meet with approval. All authors have approved the response letter and the revised version of the manuscript. 

And due to our writing ability, understanding and logic are limited and we look forward to your generosity. Due to the time limit of our project and the need to employ professional title, and this paper is very important to our team, so we sincerely hope that you can recommend our revised version of the manuscript to accept for publication in the journal Plos One as soon as possible. If you have any queries, please do not hesitate to contact me. 

 Thank you again for your valuable comments and suggestions. I'm looking forward to hearing from you soon in due course.

 Best regards,

 Zhangyi Wang, Master's degree, Nurse Practitioner, 283537548@qq.com, 

 Nursing Department, Affiliated Hengyang Hospital of Hunan Normal University & Hengyang Central Hospital, Hengyang, Hunan Province, 421001, China.

---

## [Editor Report · Decision Letter 1]

25 Jul 2024

Problem-solving ability and future time perspective among the Chinese nursing interns: The mediating role of future work self

PONE-D-24-16452R1

Dear Dr. Wang,

We’re pleased to inform you that your manuscript has been judged scientifically suitable for publication and will be formally accepted for publication once it meets all outstanding technical requirements.

Kind regards,

Mukhtiar Baig, Ph.D.

Academic Editor

PLOS ONE

---

## [Editor Report · Acceptance letter]

30 Jul 2024

PONE-D-24-16452R1 

PLOS ONE

Dear Dr. Wang, 

I'm pleased to inform you that your manuscript has been deemed suitable for publication in PLOS ONE. Congratulations! Your manuscript is now being handed over to our production team.

Kind regards, 

on behalf of

Professor Mukhtiar Baig 

Academic Editor

PLOS ONE